# Risk Factors Associated with Malnutrition among Children Under-Five Years in Sub-Saharan African Countries: A Scoping Review

**DOI:** 10.3390/ijerph17238782

**Published:** 2020-11-26

**Authors:** Phillips Edomwonyi Obasohan, Stephen J. Walters, Richard Jacques, Khaled Khatab

**Affiliations:** 1School of Health and Related Research (ScHARR), University of Sheffield, Sheffield S1 4DA, UK; s.j.walters@sheffield.ac.uk (S.J.W.); r.jacques@sheffield.ac.uk (R.J.); 2Department of Liberal Studies, College of Administrative and Business Studies, Niger State Polytechnic. Bida Campus, Bida 912231, Nigeria; 3Faculty of Health and Wellbeing, Sheffield Hallam University, Sheffield S10 2BP, UK; K.Khatab@shu.ac.uk

**Keywords:** malnutrition, stunting, underweight, wasting, overweight, anthropometric indices, undernutrition, overnutrition, under five, Sub-Saharan Africa

## Abstract

*Background/Purpose*: Malnutrition is a significant global public health burden with greater concern among children under five years in Sub-Saharan Africa (SSA). To effectively address the problem of malnutrition, especially in resource-scarce communities, knowing the prevalence, causes and risk factors associated with it are essential steps. This scoping review aimed to identify the existing literature that uses classical regression analysis on nationally representative health survey data sets to find the individual socioeconomic, demographic and contextual risk factors associated with malnutrition among children under five years of age in Sub-Sahara Africa (SSA). *Methods*: The electronic databases searched include EMBASE (OVID platform), PubMed (MEDLINE), Cumulative Index to Nursing and Allied Health Literature (CINAHL), Scopus, Web of Science (WoS) and Cochrane Library. Only papers written in the English language, and for which the publication date was between 1 January 1990 and 31 July 2020, were included. *Results*: A total of 229 papers were identified, of which 26 were studies that have been included in the review. The risk factors for malnutrition identified were classified as child-related, parental/household-related and community or area-related. *Conclusions*: Study-interest bias toward stunting over other anthropometric indicators of malnutrition could be addressed with a holistic research approach to equally address the various dimension of the anthropometric indicators of malnutrition in a population.

## 1. Introduction

Malnutrition is the intake of an insufficient, surplus or disproportionate amount of energy and/or nutrients [1]. Malnutrition is a significant global public health burden with greater concern among children under five years [2]. In an attempt to address this global challenge of malnutrition, the World Health Organization (WHO) member states recently signed into effect a commitment to nine global targets by 2025, including a 40% reduction in childhood stunting, a less than 5% prevalence of childhood wasting, to ensure no increase in the number of children who are overweight [3], and to end all forms of malnutrition by 2030 [3,4]. With less than five years to the target date, the progress has remained relatively slow, with no country working toward full actualization of the nine targets [5]. Though there has been considerable global decline that has been noticed in childhood stunting, there are over 150 million, 50 million and 38 million children remaining stunted, wasted and overweight, respectively [5]. However, contrary to the expectation and in line with a global target on malnutrition to keep the rate of overweight in children constant, in 2018 there were over 40 million children under five who were overweight [6], indicating a gradual global increase in overweight children. There is the possibility that the number of overweight children will increase further in the aftermath of covid-19 global lockdown. Just as most countries are witnessing multiple forms of malnutrition indices, in the same way, individual children are found to suffer from two or more forms of malnutrition indicators globally [5].

In recent times, Sub-Saharan Africa (SSA) has had so much to grapple with in terms of the malnutrition burden. In 2015, SSA accounted for more than 30% of global underweight children [7]. Additionally, in 2018, despite a worldwide decline in childhood stunting, the African region witnessed a rise in the relative figure from 50.3 million to 58.8 million children [6]. Interestingly, the 7.1% prevalence of children under five wasting in Africa is lower than the global rate of 7.3% [8]. Within the SSA region, sub-regional variations in malnutrition are reported in the literature. Akombi et al. [9] concluded in their study that countries in East and West Africa bore the greatest burden of malnutrition in the SSA region. Malnutrition is expressed through either undernutrition (the most common in less developed countries), a situation of low protein-energy intake [10] (which usually manifests at different anthropometric indices in stunting, wasting and underweight), and/or overnutrition, which is commonly associated with too great an intake of protein-energy (a situation widely associated with developed society, but of less concern in the developing countries [11], perhaps a dangerous position to assume especially in Africa).

Beside the SSA region, malnutrition has posed some serious public health challenges in other regions of the world. For instance, in Asia, though considerable steps have made towards the global target, there are lapses in achieving the targets that still exist. The region has experienced a prevalence of overweight among children under five years, which is less than the global average, but it also experienced more than the global average in stunting and wasting, which stood at 22.7% and 9.4%, respectively [5]. Similarly, the Latin American region has in the last three decades been working to deal with the burden of malnutrition, and is yet to achieve significant progress in some parts of the region. UNICEF’s 2019 report states that almost 20% of children under five in Latin America and the Caribbean were either suffering from any of the indices of malnutrition or overlapping in any two of them [12]. Galasco and Wagstaff stated that by 2030, and with the current space for an annual reduction rate in stunting, Brazil, Costa Rica, the Dominican Republic and Mexico are on course to reduce the stunting rate by 50% [13]. Overnutrition is a burden in most developed regions of the world. In 2017, more than a quarter of children in more than 80% of the states in America were either overweight or obese due to inconsistent access to good food. The public health outcomes of malnutrition, manifesting in stunting, wasting, underweight, marasmus, kwashiorkor, edema and perhaps death, are functions of macronutrients and micronutrients missing from the child’s meal [14].

Generally, malnutrition can lead to cognitive and physical impairment in children, especially those under five years old, with a high rate of morbidity and mortality [15,16]. A child’s fundamental right to a higher level of physical and mental health development worldwide is boosted with access to good nutrition [13]. Martinez and Fernandez identified three analytical areas of concerns in addressing the burden of malnutrition. First is the analyses of the capacity of any society to be self-sufficient in terms of food security for all. Secondly, they look at how variations in the demographic and epidemiological set-up have affected the nutrition status of the population, and thirdly, they look at how the life-style of the people has affected their nutrition status [13]. To effectively address the problem of malnutrition, especially in resource-scarce communities, knowing the prevalence, causes and risk factors associated with it are essential steps. This review is part of a doctoral degree work on multi-morbidities in children of under five years in Nigeria. Studies that have addressed malnutrition in Nigeria with a nationally representative sample are few, and this has necessitated a broader coverage in this scoping review to other areas with similar socio-economic and demographic set-ups as in SSA. Additionally, the methodology involved in the scoping review includes qualitatively reviewing the content of study, with a view to identifying the study gaps in the outcomes of interest, the analytical methods and the study population, which have all influenced the use of a scoping review in this study.

### The Aim of the Scoping Review

This scoping review aimed to identify existing literature that used classical regression analysis, (analysis that is based on frequentist statistics), on nationally representative health survey data sets to find the individual socioeconomic, demographic and contextual risk factors associated with malnutrition among children under five years of age in Sub-Sahara Africa (SSA).

## 2. Methodology

### 2.1. Design

The methodological pattern used in this scoping review followed Arkey and O’Malley [17], Lecac et al. [18], and the Agency for Healthcare Research and Quality (AHRQ) [19]-enhanced framework, recommendations and guidelines, respectively. The steps include the following: (1) identify the research question, (2) identify the relevant study sources, (3) select sources of evidence and eligibility criteria, and (4) chart data [20]. However, the pattern of reporting the results in this scoping review follows the Preferred Reporting Items for Systematic Reviews and Meta-Analyses extension for Scoping Reviews (PRISMA-ScR) guidelines [21,22].

### 2.2. Protocol and Registration Declaration

There was no review protocol and registration done for this scoping review.

### 2.3. Identification of the Research Questions

The research question was stated having been guided by PICOTS (population, intervention, comparators, outcomes, timing and study design) framework of Agency for Healthcare Research and Quality (AHRQ) [19]. 

The primary research question for this scoping review is what risk factors are associated with the malnutrition status of children less than five years of age in Sub-Saharan Africa countries that used classical regression methods to analyze a nationally representative survey data set?

Other secondary research questions are:What are the existing examples of evidence of individual and contextual risk factors associated with the malnutrition status of children under five years in Sub-Saharan Africa countries?What evidence exists in the use of classical regression analysis methods to determine the risk factors related to the malnutrition of children under five years in Sub-Saharan African countries?

### 2.4. Eligibility Criteria

The studies included in the review followed the PICOTS (population, interventions, comparators, outcomes, timing and study design) criteria enumerated and defined in Table 1 below.

### 2.5. Identify the Relevant Sources of Evidence

#### Information Sources

The first author (PEO) of the School of Health and Related Research (ScHARR), the University of Sheffield, United Kingdom, carried out the literature search. The process was done at least twice on each of the databases consulted and we compared the outcomes to ensure that relevant papers were not excluded. The selection of bibliographies for screening was done on the basis of keywords and subject headings. The electronic databases searched include EMBASE (OVID platform), PubMed (MEDLINE), Cumulative Index to Nursing and Allied Health Literature (CINAHL), Scopus, Web of Science (WoS) and Cochrane Library. Only papers written in the English language, and published between 1 January 1990 and 31 July 2020, were included.

### 2.6. Selection of Sources for Evidence and Eligibility Criteria

#### Search Strategy

In this scoping review the search strategy involved searching for key terms or text words individually. The phrases were first searched in EMBASE (OVID platform) using “map terms to subject heading”. The search terms applied were derived from the PICOTS categories and they include the variants of Sub-Saharan Africa, under five years, the determinants or risk factors, malnutrition status, and (with/without) regression techniques. These various terms were used with appropriate Boolean connectors, ‘AND/OR’, and with publication dates and research designs applied as restrictions. The sample of the search strategy in EMBASE is displayed in Table 2 below.

In the EMBASE search strategy result (Table 2), the publication period was set as ‘limit to last 30 years’, (because the default search time was set at 1974 to July 2020). However, for other electronic databases, the publication period was restricted to between 1990 and 2020. The timing was informed over the periods when (i) Demographics and Health Surveys had been conducted in Nigeria, (ii) the UNICEF conceptual framework on causes of malnutrition began, (iii) the Millennium Development Goals were in effect, (iv) the WHO nine targets for malnutrition were on course, and (v) the Sustainable Development Goals were in progress. The search was conducted in the last week of July 2020.

### 2.7. Selection Process

The reviewer, PEO, screened all the selected literature for titles and abstracts using the inclusion and extraction criteria as a benchmark (Table 1). This process was also done twice in two citation managers platforms (Endnote and Zotero). Any discrepancy observed was resolved by examining them more closely. A full-text reading was conducted for all the selected articles. Papers excluded were noted with reasons. Three overseeing team members vetted this process.

### 2.8. Data Charting Management

Initially the data extracted from the included articles were deposited into a Microsoft Excel spreadsheet designed by the reviewer specifically for this review. The relevant information obtained includes authors/year of publication, the survey type, the sample size, classical regression type and country of study. Other information includes the study aim, the outcomes (malnutrition status), the prevalence, various predictor variables assessed (child-related variables, parental/household-related variables and contextual or community-related variables), significant risk factors found for each of the malnutrition-related indicators, the specific conclusion reached, and the statistical software used for computation.

## 3. Results

The results section reports the profile of the quantitative analysis of risk factors associated with malnutrition in under five children in SSA following the Preferred Reporting Items for Systematic Reviews and Meta-Analyses extension for Scoping Reviews (PRISMA-ScR) checklists [21,22]. 

### 3.1. Selection of Sources of Evidence

Figure 1 represents the flowchart of the included studies. A total of 224 unique papers were identified from the various electronic databases (EMBASE = 12, PUBMED = 18, WOS = 74, Scopus = 103, Cochran Library = 0, CINAHL = 12). Additionally, five other studies were retrieved from others sources (the reviewer’s files). 

Twenty-five studies were duplicated in the search at different times (twice, thrice, four or five times). The duplication led to the removal of 47 titles. Out of a total of 177 studies screened for titles and abstract, 138 studies were removed for not meeting the inclusion criteria. A total of 26 studies were finally selected for this study after excluding 13 papers. The reasons for excluding these papers are listed in the chart above (Figure 1).

### 3.2. Characteristics of Sources of Evidence

To answer the questions raised in this scoping review, the relevant information was extracted from the selected papers and is presented in Table 3 and Table 4. This section describes the characteristics of the sources of evidence.

#### 3.2.1. Characteristics of Study Setting

Table 3 includes elements of the study setting. The unit of analysis in this scoping review is the country of study. Though there were 26 articles selected in this review, two studies (Kennedy et al., 2006 and Ntoimo et al., 2014) analyzed the data separately for three countries each, resulting in risk factor estimates for 30 country unique studies (and16 unique countries). The highest number of publications came from Nigeria, having five studies representing 16.7% [15,23,24,25,26], followed by Ethiopia [27,28,29,30], and articles with multi-countries [31,32,33,34] have four studies each. The multi-country articles are studies that focused on more than one country, with the countries’ data sets pooled together and analyzed as one study. Ten countries (Swaziland, Senegal, Rwanda, Malawi, Kenya, Ghana, Equatorial Guinea, Democratic Republic of Congo (DRC), Cameroon, and Central Africa Republic (CAR)) had one study each.

#### 3.2.2. Characteristics of Study Analytical Methods

One of the inclusion criteria for this scoping review was that the statistical analytical techniques must be classical statistical regression methods. Table 3 contains the listing of various statistical analysis techniques used for each study. The most frequently used technique was logistic regression (LR). There were 21 studies (70%) out of the 30 selected country-based studies that used one form of LR or another (multivariate LR, multiple LR, ordinal LR or conditional LR). Five studies applied multilevel regression analysis, two studies used multinomial regression analysis and two other studies, including Aheto [36], used a relatively unpopular statistical approach, Simultaneous Quantile Regression (SQR), a technique used in modeling regression concerning quantiles (or percentiles) instead of the usual modeling about the mean (mean regression), while Takele et al. [30] used a Generalized Linear Mixed Model (GLMM).

#### 3.2.3. Characteristics of Study Outcomes

In Table 4, it was observed that the most studied outcome was stunting. It was the focus of 28 (representing 93.3%) out of the 30 country-based articles (with stunting appearing in 16 publications as the only outcome variable and 12 studies paired with other malnutrition indicators). Wasting and underweight appeared in 13 reports, while overweight was only included in two papers. Furthermore, undernutrition (stunting, wasting and undernutrition) was the outcome of interest in six studies. However, there was only one study that focused on all the four indicators of malnutrition (stunting, wasting, underweight and overweight) [45]. 

#### 3.2.4. Characteristics of Significant Risk Factors

Table 4 also contains the list of predictor variables considered for each study selected for this scoping review. It lists the significant risk factors concerning stunting, wasting, underweight and overweight of children less than five years old. The choice of predictor variables studied in some of the articles selected was guided by the UNICEF framework of causes of undernutrition in children [46]. These were classified as child-related (CR), parental/household-related (PHR) and community- or area-related factors (AR). 

Among the child-related risk factors, gender and age (in months categories) were the most frequent significant predictors of stunting (13 studies), wasting (four reports), underweight (4 studies), overweight (no study) and stunting (12 articles), wasting (six reports), underweight (4 studies) and overweight (1 study), respectively. In the parental category, maternal education was the most active predictor in 14, 3, 5 and 1 studies for stunting, wasting, underweight and overweight, respectively. Out of the 28 studies that investigated stunting, 16 reported a significant association of household wealth status with stunting. Place of residence from the community-related category was significant in stunting (five studies), wasting (three studies) and underweight (one study). Significant comorbidity was found for a child having diarrhea in the last two weeks before the survey with stunting (four studies) and underweight (two studies) captured in this review.

## 4. Discussion

This scoping review aimed to identify the existing literature that used classical regression analysis on nationally representative health survey data sets to find the individual socioeconomic, demographic and contextual risk factors associated with malnutrition among children under five years of age in Sub-Sahara Africa (SSA). The review identified 26 studies and the risk factors for malnutrition, which were classified as child-related, parental/household-related and community or area-related factors. The risk factors for malnutrition identified included age, gender, comorbidities (such as diarrhoea), maternal education, household wealth and place of residence.

This scoping review has demonstrated the importance researchers have attached to studying malnutrition (especially in children under five years) in order to provide a basis for evidence-based decision-making toward meeting the WHO’s nine targets on malnutrition by 2025. Some of the most common determinants of malnutrition indicators include child’s age, sex, birth size, breastfeeding status, and whether the child had a fever in the last two weeks before the survey. Other indicators are the mother’s age, education level, Body Mass Index, and father’s education level. In the household category, wealth status, number of children under five years in the household, source of information, and improved building materials, and from the community-related category, place and region of residence, and Gross Domestic Product (GDP). However, there are a few issues from these studies that need to be discussed here. 

Firstly, malnutrition in children is a situation where children are either undernourished (less necessary energy and nutrient intake) or ‘over-nourished’ (too much necessary energy and nutrient intake) [1]. The authors believed that ‘malnutrition’ and ‘malnourished’ are two different things. Malnourishment (or undernourished or undernutrition) is a component of malnutrition. However, most studies often show some inconsistencies in the classification of malnutrition in this direction. The anthropometric indices generally used by the World Health Organization to measure nutritional status stipulate height-for-age, weight-for-height and weight-for-age for measuring stunting, wasting and underweight, respectively. These indices are computed as ‘standard deviation units (Z-scores) from the median of the reference population’ [47]. In the 2018 NDHS, for instance, malnutrition was classified into four areas, as follows: (i) stunting in a child too short for his/her age with a height-for-age Z-score less than minus two standard deviations (−2SD) from the median; (ii) wasting in a child is acute undernutrition status, which describes a child’s status whose weight-for-height Z-score is less than minus two standard deviations (−2SD) from the median; (iii) underweight is a composite extraction of both stunting and wasting, giving a weight-for-age Z-score of below minus two standard deviations (−2SD) from the median; and (iv) overweight, in this case, refers to a child whose weight-for-height Z-score is above two standard deviations (+2SD) from the median of the reference population [47]. So, most studies that focused on malnutrition have always considered stunting, wasting and underweight as only proxies for nutritional status without including overweight [48]. Some of these studies that have excluded overweight in their nutritional status often used the word ‘undernutrition’, while others used ‘malnutrition’, and some used the terms interchangeably [48,49]. The argument here surrounds the exclusion of overweight when determining the nutritional or malnutrition status of children in a population. Magadi et al. [32] reported that overweight was excluded from among the malnutrition indicators because it is not of greater importance in the least developed countries. This measure of excluding overweight in effect can lead to underestimating the nutrition status of the population under study. In a recent paper, WHO grouped malnutrition into three essential areas, as follows: undernutrition, micronutrient deficiency and overweight related malnutrition [1]. Undernutrition involves not getting the adequate nutrients necessary for daily activities, while overnutrition is getting more nutrients than you can utilize daily [50]. So, malnutrition is a composite of undernutrition and overnutrition [49]; as such, we submit that overweight should always be included when determining the malnutrition status. In our opinion, the reasons why researchers often exclude overweight in nutritional (or malnutrition) status is that the analysis involves some statistical manipulations, and the fact that overweight’s anthropometric measures obviously connect with those of wasting. Resolving the problem in computation is done by including overweight into the application of ‘Composite Index of Anthropometric Failure (CIAF)’ [51], or by simple use of ‘composite index’ computation [52].

The second issue of concern from some of the studies in the scoping review is in the attention given to stunting over other anthropometric indices of malnutrition. This scoping review identified that for every ten studies on malnutrition, at least nine studies are investigating stunting. This trend in studying stunting may be related to the need to meet the WHO target of 40% reduction in stunting prevalence by 2025 [5], and stunting’s obvious association with poverty and hunger, which are major characteristics of the least developed and war/conflict-torn nations. These reasons, however, cannot justify the almost absence of equal attention being paid to other malnutrition indicators, especially overweight, which is seen to be increasing in some populations [53], and may increase further in the aftermath of covid-19 global lockdown. 

The third issue of concern is the multiple overlaps in the malnutrition indicators. Though few studies have focused on two or more anthropometric indicators of malnutrition, they were analyzed individually using classical logistic regression methods. In some populations, there are tendencies for multiple forms of malnutrition indicators in children [5,51,54]. Not many of the studies considered in this review evaluated the multiple overlaps in these anthropometric indices. This observation is a gap in the study. However, with appropriate statistical techniques, it becomes easy to determine the prevalence of the simultaneous occurrence of anthropometric indices among children in a population [51], thereby determining their risk factors in a population. There are over 3.6% and 1.8% children under five globally who are both stunted and wasted, and stunted and overweight, respectively [5]. However, wasting and overweight are mutually exclusive; as such, we do not expect multiple overlaps in them.

Finally, the issue of inconsistencies found in some studies concerns the proper way of categorizing undernutrition indicators (stunting, wasting and underweight) into moderate and severe undernourishment [24,32,47].

For instance, a stunted child has height-for-age (HAZ < −2SD), on a scale, a severely stunted child has HAZ < −3SD. Since stunted is moderate plus severe, then the moderately stunted child is −3SD ≤ HAZ ≤ −2SD. The same classification holds for other anthropometric indicators for undernutrition as displayed in the chart above (Figure 2).

## 5. Strengths and Limitations

This scoping review has some level of strengths. (i) This study is about the first scoping review on risk factors associated with malnutrition in children under five in SSA countries that used classical statistical regression modeling techniques on nationally representative survey samples. (ii) The identification of some grey areas that urgently need research cover, especially in the field of using appropriate statistical methods that will compositely determine the actual index of malnutrition in a population. However, there are some limitations, which include but are not restricted to the following: (i) Some potential studies may have been excluded due to the search strategies adopted. (ii) The grey literature search to seek for possible papers was not done. (iii) The references of the included publications were not searched through to ascertain more pieces of evidence. (iv) SSA countries include other countries that are not English-speaking, so some potential papers not written in English from these countries may have been lost to our search. (v) The studies included had analytical techniques restricted to classical statistics regression methods (analysis based on frequentist statistical methods); therefore, potential papers that used Bayesian statistical methods in their analyses were excluded. (vi) Linear regression as an analytic technique was omitted in the search and this may have excluded some potential papers. (vii) There was no assessment of the potential risk or publication bias conducted.

## 6. Future Work

Areas not covered in this review, especially to satisfy the limitations highlighted above, are potential work for future studies. More important is a review that will map out a piece of study evidence on malnutrition that used either classical regression analysis or Bayesian analysis methods, or both. In addition, studies that include overweight and/or micronutrient deficiencies as part of the indicators of malnutrition among children under five years are urgently needed. Furthermore, studies that will explore the interrelationship between malnutrition and other childhood diseases using appropriate statistical techniques while recognizing the interdependencies of these diseases are areas of future interest.

## 7. Conclusions

In this scoping review, we have identified several significant risk factors that predict the probability that a child under five years of age in an SSA country will develop malnutrition status. These factors were classified as child-related (CR), parental/household-related (PHR) and area-related (AR) variables. The CR include child’s age, sex, birth weight, type of birth, birth type, diarrhoeal, and place delivered. Factors related to parental/household include mother’s education, breastfeeding, BMI, birth interval, mother’s health-seeking status, mother’s age, household wealth status, improved sanitation, number of children under 5 years in the household, maternal health insurance, type of toilet facilities and cooking fuel, while among the area-related (AR) variables were forest cover lost, community region, and community illiteracy rate. To prevent the wide spread of malnutrition in developing countries, these significant risk factors must be taken into consideration when developing practice and policy formulation. Central to these controls are the maternal education and health status. Pregnant and nursing mothers should have access to a balanced diet.

The review also discovered that there was a study-interest bias toward stunting as an index over other anthropometric indicators of malnutrition. Furthermore, the review also identified some limitations in the current studies reviewed when overweight and/or micronutrient deficiencies were excluded as indices of malnutrition. In the authors’ opinion, the exclusion may be partly related to the methodological complications involved in determining the true status of malnutrition when these indices are included. Some of the nationally representative surveys used in the studies reviewed collected information regarding the overweight and/or micronutrient status of children under five years. Micronutrient deficiencies in children of under five years in developing countries are measured by the levels of iron, iodine and vitamin A intake [55]. Apart from iron, which was measured through a biomarker examination of blood samples to establish the anaemia status, iodine and vitamin A were determined subjectively through examining the nature of the foods the child consumed a day before the survey [47]. This cannot give an objective assessment of the status of the micronutrients present in a child. As such, researchers often find it difficult to include them while determining the true malnutrition status of children under five years old in developing countries. In addition, the review identified some inconsistences in the sub classifications of the malnutrition indicators into severe, moderate and mild, while applying the WHO anthropometric cut off points.

Finally, barely five years to the set date of achieving the WHO’s nine targets of malnutrition in children, in this scoping review we conclude that a holistic research approach to equally address the various dimensions of anthropometric indicators of malnutrition in a population is needed. Evidence from such findings will be valuable documents in the hands of many planners/policymakers for informed decision making.

## Figures and Tables

**Figure 1 ijerph-17-08782-f001:**
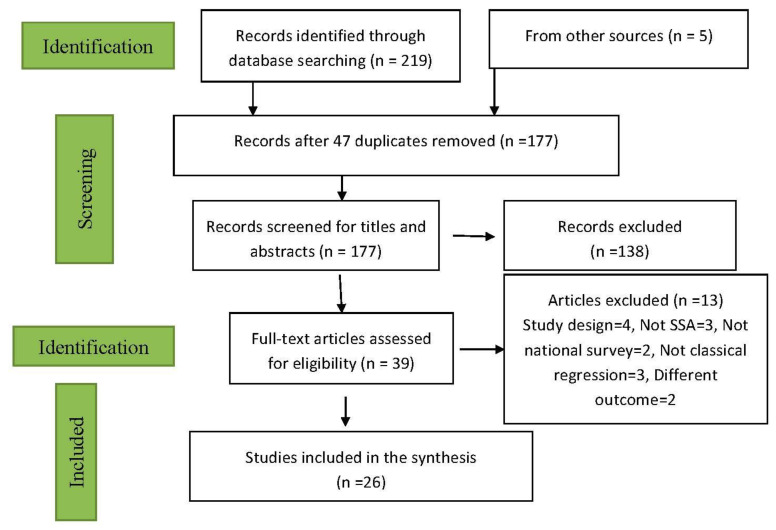
Flowchart of inclusion of studies for malnutrition review.

**Figure 2 ijerph-17-08782-f002:**
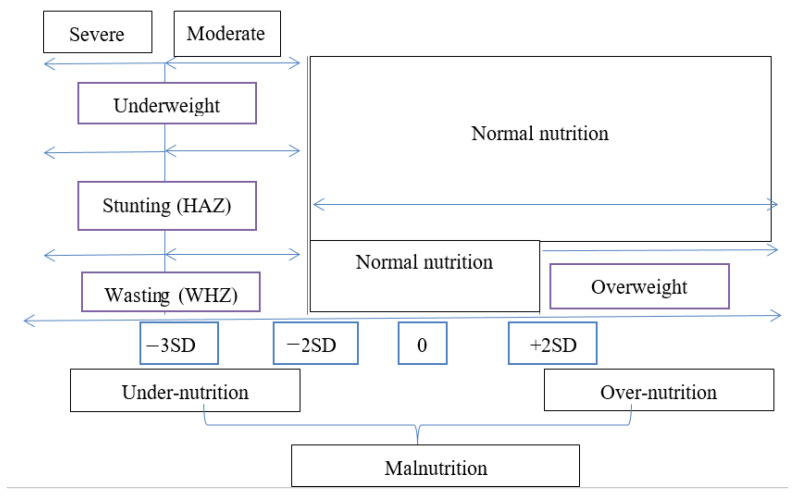
Showing the classifications of Anthropometric indicators of malnutrition.

**Table 1 ijerph-17-08782-t001:** Structure for eligibility criteria in malnutrition studies.

Criteria	Determinants	Inclusion Criteria	Exclusion Criteria
Population (P)	Children Under five years are those less than five years of age.	The studies included both male and female children less than five years of age and residing in any of the Sub-Saharan Africa (SSA) countries. We also include publications that involved both adults, children above five years and children under five years; the provided data for under five years were reported differently from others.	Studies involving older children, but no separate reporting for data involving under five years were taken
Intervention (I)	Risk factors associated with malnutrition classified as child-related variables, parental/care-givers-related variables, household-related socioeconomic status, demographic status and area characteristics.	Studies that focused on predictors or risk factors or determinants of malnutrition among under five or pre-school children in SSA that covered both individual and contextual exposures.	
Comparator (C)	These studies involved two mutually exclusive groups: those that are ‘nourished’ and ‘malnourished’ for which we compared the exposures.	However, we included studies which declassified malnutrition status into stunting, wasting, underweight, overweight and nutrition status.	
Outcomes (O)	The main outcome is the **malnutrition status (MNS)** of children under five years. The MNS is determined through the measurement of anthropometric indices expressed as either stunting (assessed through height-for-age), wasting (assessed through weight-for-height) underweight (assessed through weight-for-age) each with Z-score ≤ −2SD from the median of the reference population, and overweight (estimated via weight-for-height) with Z-score > +2SD from the median of the reference population.	Studies that used any of the indicators or composite index of stunting, wasting, underweight and/or overweight, were onsidered for inclusion, as well as studies that used severity level (such as acute, mild etc.) for the indicators to classify malnutrition or nutrition status, and as such only one aspect was chosen (mild, severe or acute).	
Timing (T)	The time articles were published.	The publication period for the article is between 1st January 1990 and 30th July 2020 to capture recent publications on the topic from when UNICEF’s conceptual framework of causes of malnutrition was in effect, the MDG and SDGs	All papers published outside the period 1990–2020.
Settings/Design (S)	These studies must be a nationally representative health-related survey in one or more of the Sub-Saharan African countries. These include Demographic and Health Survey (DHS), Multiple Indicator and Cluster Survey (MICS), AIDS Indicator Survey (AIDSIS), and any other countries’ specific survey with a national spread.	Observational studies such as cross-sectional studies that focused on risk factors as predictors. The search also included those studies that applied classical regression for the analyses.	Studies that used other methods such as Bayesian or spatial analyses techniques. Papers on nationally representative health surveys not conducted for SSA countries. Community-based, facility-based and non-nationally representative surveys. Systematic review studies.

**Table 2 ijerph-17-08782-t002:** Draft search strategy and terms for EMBASE (OVID).

S/N	Terms and Keywords	Results
1	Demographic and health survey OR DHS OR AIDS indicator survey OR malaria Indicator survey OR multiple indicator cluster survey OR health survey OR Nutrition Survey OR MIS OR MICS	258,604
2	Sub-Saharan Africa OR SSA	37,452
3	1 AND 2	1764
4	Socioeconomic OR demographic OR contextual OR environmental OR community OR determinants OR risk factor OR predictor	3,298,141
5	Malnutrition OR stunting OR wasting OR underweight OR under-weight OR overweight OR over-weight OR Nutrition Status OR Nutritional	451,419
6	4 AND 5	107,110
7	3 AND 6	134
8	Limit 7 to human and English language and infant < to one year > OR preschool child < 1 to 6 years >	40
9	Logistic regression OR multilevel regression OR multinomial logistic OR random-effects OR hierarchical OR fixed effects	55,708
10	8 AND 9	12
11	Limit to last 30 years (1990 to 2020)	12

**Table 3 ijerph-17-08782-t003:** Characteristics of the 26 studies included in the review/synthesis.

Author and Data	Country	Study Design	Participants (N) and Study Population	Analysis Methods	Software Used
Adekanmbi et al. (2013) [23]	Nigeria	2008 Nigeria Demographic and Health Survey (NDHS)	28,647 0–59 months	Multilevel logistic regression	Stata
Acharya et al. (2020) [31]	Multi-countries	Demographic and Health Survey and Global Forest Change dataset	Women 15–49 years (25,285) and 12–59 months (73,941)	Logistic regression methods	Stata
Agadjanian et al. (2003) [35]	Angola	1996 Angola Multiple Indicator Cluster Survey (AMICS)	Number of participants not stated 6–59 months	Multivariate logistic regression	Stata
Aheto (2020) [36]	Ghana	2014 Ghana Demographic and Health Survey (GDHS)	2716 Under five years	Multivariate Simultaneous quantile regression	R-Package
Akombi et al. (2019) [24]	Nigeria	2003–2013 NDHS	22,217 0–59 months	Logistic regression	Stata
Akombi et al. (2017) [15]	Nigeria	2013 NDHS	24,529 0–59 months	Multilevel logistic regression	Stata
Amaral et al. (2017) [37]	Uganda	Uganda National Panel Survey (UNPS)	3427 under-5 years	Binary logistic regression	Stata
Amare et al. (2019) [27]	Ethiopia	2016 Ethiopia Demographic and Health Survey (EDHS)	9419 under-5 years	Multiple logistic regression	Stata
Custodio et al. (2008) [38]	Equatorial Guinea	2004 nationally survey	552 Under five years	Multivariate logistic regression	PEPI
Doctor & Nkhana-Salimu (2017) [7]	Malawi	1992–2016 Malawi Demographic and Health Survey (MDHS)	31,630 Under five years	Logistic regression	Nil
Gebru et al. (2019) [28]	Ethiopia	2016 Ethiopia Demographic and Health Survey (EDHS)	8855 Under five years	Multilevel logistic regression	Stata
Kennedy et al. (2006) [39] Multi-countries	Angola	2001 Angola Multiple Indicator Cluster Survey (AMICS)	5116 Under five years	Logistic regression	SPSS
	Central African Republic	2000 Multiple Indicator Cluster Survey (CARMICS)	12,499 Under five years	Logistic regression	SPSS
	Senegal	2000 Multiple Indicator Cluster Survey (SMICS)	8319 Under five years	Logistic regression	SPSS
Kuche et al. (2020) [29]	Ethiopia	2016 Sustainable Undernutrition Reduction in Ethiopia (SURE)	1848 6–23 months	Ordinal logistic/linear regression model	Nil
Machisa et al. (2013) [40]	Swaziland	2008–2007 Swaziland Demographic and Health Survey (SDHS)	1155 6–36 months	Multinomial logistic regression	Stata
Magadi (2011) [32]	multi-countries	2003–2008 Demographic and Health Survey (DHS)	55,749 Under five years	Multilevel logistic regression	MlwiN
McKenna et al. (2019) [11]	Democratic Republic of Congo	2013–2014 Democratic Republic of Congo Demographic and Health Survey (CDHS)	3722 6–59 months	Logistic regression	SPSS
Miller et al. (2007) [41]	Botswana	2000 Botswana Multiple Indicator Cluster Survey (BMICS)	2723 Under five years	Multilevel logistic regression	MlwiN
Nankinga et al. (2019) [42]	Uganda	2016 Uganda Demographic and Health Survey (UDHS)	3531 under-5 years	Multivariate logistic regression	Stata
Nshimyiryo et al. (2019) [43]	Rwanda	2014–2015 Rwanda Demographic and Health Survey (RDHS)	3594 Under five years	Logistic regression	Stata
Ntoimo et al. (2014) [25] multi-countries	Cameroon	2011 Cameroon Demographic and Health Survey (CDHS)	5053 Under five years	Logistic regression	Nil
	Nigeria	2008 Nigeria Demographic and Health Survey (NDHS)	18,823 Under five years	Logistic regression	Nil
	Democratic Republic of Congo (DRC)	2007 Congo Demographic and Health Survey (CDHS)	3777 Under five years	Logistic regression	Nil
Ssentongo et al. (2019)	Uganda	2015–2016 Uganda Demographic and Health Survey (UDHS)	4765 0–5 years	Logistic regression	Nil
Takele et al. (2019) [30]	Ethiopia	2016 Ethiopia Demographic and Health Survey (EDHS)	8743 Under five years	Generalized Linear Mixed Model	Nil
Tusting et al. (2020) [33]	SSA countries	Demographic and Health Survey (DHS), Malaria Indicator Survey (MIS) and AIDS Indicator Survey (AIDSIS)	824,694 0–5 years	Conditional logistic regression	Nil
Mishra et al. (2007) [44]	Kenya	2003 Kenya Demographic and Health Survey (KDHS)	2756 0–4 years	Logistic regression	Nil
Ukwuani & Suchindran (2003) [26]	Nigeria	1990 NDHS	5331 0–59 months	Ordinal logistic analysis	
Yaya et al. (2019) [34]	SSA countries	Demographic and Health Survey (DHS)	299,065 Under five years	Multinomial and logistic regression	Stata

**Table 4 ijerph-17-08782-t004:** Characteristics of outcomes of interest.

Author and Date	Aim of the Study	Outcome Variables Studied	Prevalence	Predictors Considered in the Study	Significant Risk Factors Identified (Stunting)	Significant Risk Factors Identified (Wasting)	Significant Risk Factors Identified (Underweight)	Significant Risk Factors Identified (Overweight)	Conclusion
Adekambi et al. (2013) [23]	To determine the predictor of childhood stunting	Stunting	25.6%	Child’s age sex, birth weight, type of birth; mother’s age, education, breastfeeding, immunization, BMI work status, birth interval, household under five size, ethnicity, mother health-seeking, type of family, wealth status; community place of residence, region, poverty rate, illiteracy rate proper sanitation and safe water	(CR): Child’s age, sex, birth weight, type of birth; (PHR): mother’s, education, breastfeeding, BMI, birth interval, mother health-seeking, wealth status; (AR): community region, illiteracy rate.	Nil	Nil	Nil	The study shows the importance of both individual and community-related risk factors in determining childhood stunting in Nigeria
Acharya et al. (2020) [31]	To establish the effect of deforestation on the individual- and household-level double burden of malnutrition in 15 SSA countries	Stunting and overweight	2.7%	Forest cover loss, child’s age, sex, mother’s education level, age, anaemia status, overweight status, household wealth, size, improve water, sanitation, own agriculture, own livestock, place of residence, a distance of cluster to the nearest road (Km)	(CR): child’s age in months, and child’s age square, (PHR): mother’s education, age wealth status, improved sanitation, (AR): forest cover lost			Forest cover lost, mother’s education, age wealth status, improved sanitation, child’s age in months, and child’s age square	
Agadjanian et al. (2003) [35]	To determine if regional or ethnic differences exist in malnutrition levels	Wasting and stunting	Nil	Place of residence, degree of war, region of residence, language spoken at home, age, full immunization for age	(CR): age, sex, immunization status, (PHR): sex of household head, mean of education of adults, ownership of radio, drinking water, language spoken	(PHR): age, mean years of schooling of adults, and language spoken	Nil	Nil	Malnutrition rates are higher than most SSA countries
Aheto (2020) [36]	To identify risk factors of under five severe stunting	Wevere stunting	5.30%	Type of birth, sex, age, had diarrhoea, had a fever, place of delivery, size at birth, number of children, health insurance, currently breastfeeding, wealth status, maternal education	(CR): birth type, age, sex, diarrhoeal, place delivered, birth size, (PHR): maternal age, and education. Numbers of children <5 years in the household, maternal health insurance, wealth status	Nil	Nil	Nil	Use of Simultaneous Quartile Regression (SQR) can benefit in addressing under 5 stunting
Akombi et al. (2019) [24]	To examine the trend and determinants of child undernutrition	Undernutrition (Stunting, wasting and underweight)		Child’s age, mother’s age sex of child, mother’s education, father’s education, wealth index, place of residence, region.	(CR): child’s age, Sex of child; (AR): maternal, place of residence, zone.	(CR): child’s age, sex of the child,	(CR): child’s age, sex of the child; (PHR): father’s education, wealth index,		
Akombi et al. (2017) [15]	To determine the associated risk factors of wasting and undernutrition	Wasting and underweight	18% and 29%	Place of residence, region, wealth index, mother work status, education, father’s education, occupation, marital status, mother’s literacy, source of drinking water, media factors newspaper, radio, television, Mother’s age, age at birth, type, mode and place of delivery, ANC, the timing of postnatal check, breastfeeding, child’s birth order, birth interval, sex, birth size, age, had diarrhoea, had a fever		(CR): child’s birth interval, sex, had a fever (PHR): place of residence, region, education, father’s education, television	(CR): duration breastfeeding, child’s sex, birth size, had diarrhoea, had a fever (PHR): the region, mother’s education, father’s education, current		
Amaral et al. (2017) [37]	to establish that greater staple food concentrations affect stunting and wasting	Stunting and wasting	Stunting (22.2%), wasting (3.1%)	Staple Budget Share, spending, place of residence, mother present, sex household head educated	(PHR): Staple Budget Share, spending, place of residence, mother present; (CR): sex of the child	(PHR): Staple Budget Share, household head educated; (CR): sex of the child	Nil	Nil	Nutritious staple food are strongly associated with higher odds of stunting and wasting
Amare et al. (2019) [27]	To establish the determinants of malnutrition among children under age 5 in Ethiopia	Stunting and wasting	Nil	Child’s age, sex. Birth order, birth weight. Mother’s marital status, age at child’s birth, educational status, BMI, working status, maternal stature. Place of residence, region, wealth status, improve drinking water, toilet type, cooking fuel type	(CR): age, sex, birth weight; (PHR): mother above primary education, BMI, stature, household wealth above poorer, type of toilet facilities and cooking fuel	(CR): Child’s age is 2years+, sex, birth weight > average; (PHR): mother’s BMI, wealth status >middle quintile	Nil	Nil	A multi-sectoral and multidimensional approach is needed to curtail malnutrition in Ethiopia
Custodio et al. (2008) [38]	To determine the underlying factors affecting the malnutrition status of children in Equatorial Guinea	Stunting	35.20%	Socioeconomic status or wealth status, household social index, and community endowment index	(CR): child’s age, (PHR): fishing by household, hospital as close at the health facility	Nil	Nil	Nil	An integrated strategy of combating poverty and improving maternal education to solve stunting problem in Equatorial Guinea
Doctor and Nkhana-Salimu (2017) [7]	To understand the trend and effect of determinants of child nutrition among Malawian children under five	Stunting and underweight	32.60%	Place and region of residence, wealth index, source of drinking water, toilet facilities, mother’s education status, age, number of under 5, child’s sex, age, birth-order, size at birth, had diarrhoea, had a fever, had a cough	(PHR): region of residence, wealth index, mother’s education status; (CR): child’s sex, age, size at birth, had diarrhoea; (Others): survey rounds	Nil	(PHR): region of residence, wealth index, mother’s education status (is Secondary+), age (is 20–30 years); (CR): child’s sex, age, size at birth, had diarrhoea, had a fever, (Others) survey round	Nil	Decline experienced in underweight and stunting among children under 5, but remain a serious public health burden in Malawi
Gebru et al. (2019) [28]	to identify individual and community-related variables associated with stunting among children in Ethiopia under 5	Stunting	38.39%	Child’s age, sex, mother’s BMI, age, education, occupation, marital status, perceived child’s birth size, the child had diarrhoea and/or fever in the last weeks, father’s education, occupation, wealth index, place of delivery, number of children under 5 in the household, antenatal care visits, mother’s age at 1st birth, birth type, birth interval and mass-media exposure.	(CR): Child’s age, sex, perceived child’s birth size, the child had diarrhoea and/or fever in the last weeks, birth type, and birth interval; (PHR): mother’s BMI, education, occupation, marital status, father’s occupation, wealth index, number of children under 5 in the household	Nil	Nil	Nil	That individual and community factors are important determinants of stunting in Ethiopia
Kennedy et al. (2006) [39] Multi-countries	To examine the relationship between wealth status and childhood undernutrition	Stunting and underweight	45.2% and 20.5%	Place of residence, women with formal education household with adequate sanitation, with access to safe water, had diarrhoea, had acute respiratory infection and wealth status	(PHR): wealth status (poorest poor and middle)	Nil	(PHR): wealth status	Nil	Prevalence of undernutrition is similar for the same socio-economic status across the place of residence in developing countries.
	To examine the relationship between wealth status and childhood undernutrition	Stunting and underweight	38.9% and 24.3%	Place of residence, women with formal education, household with adequate sanitation, with access to safe water, had diarrhoea, had acute respiratory infection and wealth status	(PHR): wealth status (poorest, and middle)	Nil	(PHR): wealth status (poor and middle)	Nil	Prevalence of undernutrition is similar for the same socio-economic status across the place of residence.
	To examine the relationship between wealth status and childhood undernutrition	stunting and underweight	25.4% and 22.7%	Place of residence, women with formal education, household with adequate sanitation, with access to safe water, had diarrhoea, had acute respiratory infection and wealth status	(PHR): wealth status (poorest and middle)	Nil	(PHR): wealth status (poorest)	Nil	Prevalence of undernutrition is similar for the same socio-economic status across the place of residence.
Kuche et al. (2020) [29]	To examine the impact of sociodemographic, agricultural diversity and women’s employment variables on child’s length-for-age z-score in children 6–23 months in Ethiopia	Length-for-age (Stunting)	Nil	Child’s dietary diversity, age, sex, household wealth, maternal education, women decision-making power, paternal domestic chores, food insecurity, minimum women dietary diversity, animal source food types, fruit and vegetable types, land owned	(CR): child’s dietary diversity, age (months), age squared, sex; (PHR): household wealth, maternal education, fruit and vegetable types, land owned	Nil	Nil	Nil	Household production of fruit and vegetables can improve a child’s length-for-age
Machisa et al. (2013) [40]	To establish the association between the use of biomass fuels for household cooking and stunting in children	Stunting	27.60%	Child’s age, sex, anaemia, birth order, preceding birth interval. Birthweight, recent episode of an acute respiratory infection, diarrhoea and fever; mother’s age, BMI, highest education, iron supplement, anaemia status; household use of biomass fuel, place of residence, region, number of people in the household, wealth index	(CR): child’s age, preceding birth interval, birthweight; (PHR): household wealth index and use of biomass fuel	Nil	Nil	Nil	The study shows that stunting in children needs to be given priority in health intervention
Magadi (2011) [32]	To determine the effect of HIV/AIDS-affected household health outcomes on children under five years in SSA	Undernutrition (stunting, wasting underweight)	Nil	Household HIV status, paternal orphan, child’s age sex, multiple births, birth order, birth interval, breastfed, birth size, place of residence, mother’s age, education, single parenting, wealth status, community HIV prevalence, country HIV prevalence, GDP per capital	(CR): child’s age sex, multiple births, birth order, birth interval, breastfed, birth size; (PHR): The place of residence, mother’s age, education, single parenting, wealth status, household HIV status, paternal orphan; (AR): community HIV prevalence, GDP per capital	(CR): breastfed, birth size; (PHR): place of residence, mother’s education, wealth status, country, household HIV status; (AR): community HIV prevalence	(CR): child’s age sex, multiple births, birth order, birth interval, breastfed, birth size; (PHR): place of residence, mother’s age, education, single parenting, wealth status, household HIV status, paternal orphan, GDP per capital	Nil	The study reveals the need for integration of HIV/AIDS improvement toward the management of child nutrition services in vulnerable communities
McKenna et al. (2019) [11]	To determine the relationship between women’s decision-making power and stunting/wasting in children under five in DRC	Stunting/wasting	35.2%/9.2%	Decide over their own income. Husband’s income, own health, large household purchases, visits to family, child’s sex, age, mother’s education, age, birth interval, number of under-5 in HHs, Number people in HHs, province (region), place of residence, wealth status	(CR): child’s sex, age; (PHR): mother’s education, age, wealth status (richest), province (region), place of residence	(CR): child’s, age; (PHR): mother’s education (primary), place of residence, wealth status (richest)	Nil	Nil	Detailed studies with more relevant and contextual variables are needed to accurately determine the effects of women’s decision-making power and undernutrition in children
Miller et al. (2007) [41]	To determine if orphan-based health inequalities measured with anthropometric data exist.	Underweight	Nil	Nil	Nil	Nil	(CR): the child being orphan, child’s age; (PHR): number of dependent children, household head education, wealth index		More data and studies are needed to fully understand the processes that the orphan-based health disparities work on
Nankinga et al. (2019) [42]	To determine the association between maternal employment and the nutritional status of children under 5 in Uganda	Nutritional status (stunting, wasting, underweight)	Nil	Residence, region, wealth status, toilet type, source of drinking water, sex of household head, marital status, maternal occupation, mother’s employer, decision-making power, the distance a problem to health services, child’s sex, age, birth weight	(PHR): maternal age is 35–49 years, education level, maternal occupation; (CR): child’s birth weight, dewormed,	(PHR): region, maternal employer, (CR); child’s sex, age, birth weight	(PHR): mother’s education, employer; (CR): child’s birthweight	Nil	Flexible labor participation for women to enable them time to care for the child
Nshimyiryo et al. (2019) [43]	To identify risk factors in stunting in Rwanda	Stunting	38%	Child’s sex, age group, parity, birth weight, had diarrhoea in last two weeks; mother’s height, educational level, took parasite-controlling drugs during pregnancy, number of days of daily intake of iron tablets, breastfed in the first hour after birth and household’s wealth index, size, access to improved water, improved toilet facility, and household place of residence, region altitude	(CR): child’s sex, age group, birth weight; (PHR): mother’s height, educational level, took parasite-controlling drugs during pregnancy, and household’s wealth	Nil	Nil	Nil	Family-related factors are the major determinants of stunting in Rwanda
Ntoimo et al. (2014) [25] multi-countries	To determine the relationship between single motherhood and stunting	Stunting	32.0%	The child died, marital status, maternal education, place of residence, occupation, wealth status, sibling size, prenatal care, breastfeeding, birth interval, BMI, widowhood, other single mothers	Nil	Nil	Nil	Nil	Single motherhood is a challenge to stunting in SSA countries which can be reduced considerably when the families of the single mother are economically empowered
	To determine the relationship between single motherhood and stunting	Stunting	41%	The child died, marital status, maternal education, place of residence, occupation, wealth status, sibling size, prenatal care, breastfeeding, birth interval, BMI, widowhood, other single mothers	Nil	Nil	Nil	Nil	
	To determine the relationship between single motherhood and stunting	Stunting	44.50%	The child died, marital status, maternal education, place of residence, occupation, wealth status, sibling size, prenatal care, breastfeeding, birth interval, BMI, widowhood, other single mothers	Nil	Nil	Nil	Nil	
Ssentongo et al. (2019)	To establish the relationship between vitamin A deficiency and deficit in linear and ponderal growth	Stunting, wasting and underweight	27%, 4% and 7%	Child age, sex, birth order, vitamin A supplementation, deworming, had diarrhea, anaemia level, wealth status, mother educated, father educated, mother working, father working, iodized salt, owns the land for agriculture, owns livestock, place of residence, region	(CR): vitamin A deficiency	Nil	Nil	Nil	VAD is associated with stunting and not with wasting and underweight
Takele et al. (2019) [30]	To determine the risk factors associated with child stunting	Stunting	Nil	Child’s sex, age, birth interval, mother’s BMI, household wealth index, source of drinking water, type of toilet facility, breastfed, mother’s education level and region	(CR): child’s sex, age, age and birthweight; (PHR): mother’s BMI, household wealth index, use of internet facility, type of toilet facility, breastfed, mother’s education level and interaction terms, source of drinking water and mother’s BMI	Nil	Nil	Nil	Children whose mothers are uneducated are at higher risk of being stunted
Tusting et al. (2020) [33]	To establish that improved housing is associated with improved child health in SSA	stunting, wasting and underweight	30%, 8% and 22%	improved drinking water, improved sanitation, house built with finished materials, improved house, the household head had secondary education+; children mean age, child sex	Finished building materials, improved housing	(PHR): improved housing	(PHR): finished building materials, improved housing	Nil	Poor housing is a predictor of health outcomes related to child survival in SSA
Mishra et al. (2007) [44]	To determine the effect of the child being orphaned or fostered, and of HIV-infected parents, on nutrition status	Stunting, wasting and underweight	Nil	The child is orphaned, fostered, HIV+ parents, the mother is HIV– but no spouse, HIV status is unknown, HIV– parents	(PHR): child’s parent HIV status is unknown	(PHR): child whose parent is HIV+	(CR): child is fostered	Nil	Welfare programs should include children that are orphans, fosters, single mothers, HIV-infected parents
Ukwuani and Suchindran (2003) [26]	To establish the relationship between women’s work and child nutritional status (stunting and wasting)	Stunting and wasting	42.6% and 8.9%	Women economic activity, maternal education, paternal education, occupation, wealth index, type of marriage, religion, duration of breastfeeding, sex of the child, birth order, prenatal care, place of delivery, birth size, food supplement, immunization, had fever, had cough, had diarrhoea, source of drinking water, types of toilet, place of residence, region	(PHR): maternal education, wealth index, religion, age at 1st birth; (CR): duration of breastfeeding, sex of the child, birth order, birth size, immunization, had diarrhoea, place of residence, age	(CR): birth size, vaccination, had a fever, toilet, age of child; (PHR): religion	Nil	Nil	
Yaya et al. (2019) [34]	To establish the effect of birth spacing interval on child health outcomes	Stunting, wasting, underweight and overweight	Nil	Inter-pregnancy interval (<24 months, 24–36 months, 37–59 months and ≥60 months)	(PHR): inter-pregnancy interval (<24 months, 24–36 months (ref), 37–59 months and ≥60 months)	(PHR): inter-pregnancy interval (24–36 months (ref), ≥60 months)	(PHR): inter-pregnancy interval (<24 months, 24–36 months (ref), 37–59 months and ≥60 months)	(PHR): inter-pregnancy interval (24–36 months (ref), ≥60 months)	The study stressed the importance of promoting an inter-pregnancy interval of between 24 and 36 months to enhance child health outcomes

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
