# Peer review of "Risk Factors Associated with Malnutrition among Children Under-Five Years in Sub-Saharan African Countries: A Scoping Review"

_ijerph, 2020, doi:10.3390/ijerph17238782_

Round 1

Reviewer 1 Report

This manuscript is well written and has a good methodology. Data is clearly presented in the tables. Considering these strengths, though, as I read the manuscript, I found some areas in which I would have appreciated greater clarity.

 I would suggest the authors modify and expand the conclusions in the abstract; should be re-written in order to provide more specific information.

Additionally, You might consider expanding the conclusion section( in the manuscript lines 317-321) to more completely provide an overview of your research results. Conclusions in the main text should be elaborated; present a brief summary of the papers main points, communicate the importance of the authors idea and the subject matter.

Author Response

Malnutrition Paper Response to Reviewer 3 Comments

I would suggest the authors modify and expand the conclusions in the abstract; should be re-written in order to provide more specific information.

Response

The conclusion part of the abstract has been expanded given the word limits allowed by the journal

Point 2:

Additionally, You might consider expanding the conclusion section( in the manuscript lines 317-321) to more completely provide an overview of your research results. Conclusions in the main text should be elaborated; present a brief summary of the papers main points, communicate the importance of the authors idea and the subject matter.

Response

The conclusion in the text has been expanded to present brief summary of the papers. See lines 363 -393

Reviewer 2 Report

Malnutrition in childhood compromises the survival and long-term well-being of the entire population. Despite extensive government/NGO efforts, malnutrition burden remains high in Sub-Sahara Africa (SSA). In this manuscript, the authors conducted a scoping review to identify the individual socioeconomic, demographic and contextual risk factors associated with malnutrition among children under-five years of age in SSA. With the clock ticking on WHO deadline for Global Nutrition Targets by 2025, this study is a timely piece of work with substantial clinical and socioeconomic importance.

Overall this is a solid review. The literature are nicely cited and the conclusions are convincing. The author’s discussion about limitations of current studies (exclusion of overweight as a subcategory of malnutrition; bias towards stunting over other anthropometric indices of malnutrition) offers valuable implications for future research and policy making.  I have only the following minor points to make. 

  1. Throughout the text, some sentences are too long-winded and need to be streamlined.
  2. There are a few typos (i.e. L168 “unquie” should be “unique”; L271: “tore” should be “torn”) and grammatical errors (i.e. L188 “it observes that”; L193 “it was” should be “there was”; L204 “On” should be “Among”; L207 “On” should be “In”; L231 “is often referred to [as] a situation…”) in the manuscript. A thorough reading for grammatical and spelling correctness is strongly recommended.
  3. Punctuations are missing or misused in multiple sentences (L60, L83, L85, L89, L144, L163, L168, L194, L213, L292, L307).
  4. The page numbers are broken in the manuscript.

Author Response

Malnutrition Paper Response to Reviewer 2 Comments

  1. Throughout the text, some sentences are too long-winded and need to be streamlined.
  2. There are a few typos (i.e. L168 “unquie” should be “unique”; L271: “tore” should be “torn”) and grammatical errors (i.e. L188 “it observes that”; L193 “it was” should be “there was”; L204 “On” should be “Among”; L207 “On” should be “In”; L231 “is often referred to [as] a situation…”) in the manuscript. A thorough reading for grammatical and spelling correctness is strongly recommended.
  3. Punctuations are missing or misused in multiple sentences (L60, L83, L85, L89, L144, L163, L168, L194, L213, L292, L307).
  4. The page numbers are broken in the manuscript.

Response:

These errors have been corrected. The entire manuscript was professionally edited

Reviewer 3 Report

Review of the manuscript entitled "Risk Factors Associated with Malnutrition among Children Under-five years in Sub-Saharan African Countries: A Scoping Review".

The review topic of the association between malnutrition of Sub-Saharan children and their risk factors is very interesting. Although I make a number of suggestions for its improvement.

Introduction
It is brief and would need to be improved in terms of the extension of concepts. The definition of malnutrition should also be extended to other areas than Sub-Saharan.

There should be a change in the way that references are cited in the text, since instead of putting the references in brackets, for example [25, 26] the authors mark them as footnotes.

Likewise, and as a final contribution, the conclusions should be elaborated in depth. The authors have done a good job and it is worthwhile to put a little more care into the conclusions.

Author Response

Malnutrition Paper Response to Reviewer 1 Comments

Point 1: t is brief and would need to be improved in terms of the extension of concepts. The definition of malnutrition should also be extended to other areas than Sub-Saharan.

Response:

Introduction has been expanded with the addition of lines 65 – 95

Point 2: There should be a change in the way that references are cited in the text, since instead of putting the references in brackets, for example [25, 26] the authors mark them as footnotes.

Response:

References adjusted as appropriate in line with the journal specifications

Point 3

Likewise, and as a final contribution, the conclusions should be elaborated in depth. The authors have done a good job and it is worthwhile to put a little more care into the conclusions.

Responses

Conclusion session has been expanded with additional comments on lines 358 - 393